# A Circular Economy Approach to Restoring Soil Substrate Ameliorated by Sewage Sludge with Amendments

**DOI:** 10.3390/ijerph19095296

**Published:** 2022-04-27

**Authors:** Wiktor Halecki, Nuria Aide López-Hernández, Aleksandra Koźmińska, Krystyna Ciarkowska, Sławomir Klatka

**Affiliations:** 1Department of Land Reclamation and Environmental Development, University of Agriculture in Krakow, Al. Mickiewicza 21, 31-120 Krakow, Poland; slawomir.klatka@urk.edu.pl; 2Montecillo Campus Address, Postgraduate College of Agricultural Sciences, Mexico-Texcoco Highway, Km. 36.5, Texcoco 56230, Mexico; lopez.nuria@colpos.mx; 3Department of Botany, Physiology and Plant Protection, Faculty of Biotechnology and Horticulture, University of Agriculture in Krakow, Al. Mickiewicza 21, 31-120 Krakow, Poland; aleksandra.kozminska@urk.edu.pl; 4Department of Soil Science and Agrophysics, University of Agriculture in Krakow, Al. Mickiewicza 21, 31-120 Krakow, Poland; krystyna.ciarkowska@urk.edu.pl

**Keywords:** biochemical activity, enzymatic biomarkers, metal availability, post-mining remediation, sewage sludge amendments, substrate enrichment

## Abstract

This study examined the use of an artificial soil substrate in a mine waste reclamation area and its effect on plant metabolic functions. Research was conducted by determining the relationship between the plants’ biochemical features and the properties of plant growth medium derived from post-flotation coal waste, sewage sludge, crushed stone and fly ash on the surface of the mine waste disposal area. Trees and shrubs were established on the material and allowed to grow for eight years. The study determined that the applied plants and the naturally occurring *Taraxacum officinale* were suitable for physio-biochemical assessment, identification of derelict areas and reclamation purposes. An evaluation of a soil substrate applied to post-mining areas indicated that it was beneficial for plant growth since it activated the metabolic functions of herbaceous plants, shrubs, and trees. The study showed that soil substrate can be targeted to improve plant stress tolerance to potentially toxic elements (PTEs). These data suggest the potential for growth and slower susceptible response to Cd, Cr, Cu, Fe, Mn, Ni, Pb and Zn. It is possible that the constructed soil-substitute substrate (biosolid material) would be an effective reclamation treatment in areas where natural soil materials are polluted by PTEs. This observation may reflect a more efficient use of soil substrate released from the cycling of organic biogene pools, in accordance with the circular economy approach. In further studies related to land reclamation using sewage sludge amendments, it would be necessary to extend the research to other stress factors, such as salinity or water deficiency.

## 1. Introduction

In reclamation projects aimed at obtaining tree cover, achieving and maintaining appropriate physical and chemical conditions means combining the reclamation process with agrotechnical methods typical of the initial management of biological reclamation [1]. Reclamation with sewage sludge mixtures may be helpful for restoring previously arable wasteland, where herbaceous or woody plants could improve soil conditions for crop plants [2].

Accumulation of potentially toxic elements (PTEs) in plants has been gaining attention, as contamination has become a major concern in recent years. Arsenic (As), cadmium (Cd), chromium (Cr), copper (Cu), mercury (Hg), nickel (Ni), lead (Pb) and zinc (Zn) are listed as priority pollutants [3]. Cistus salviifolius, a plant capable of growing in contaminated sites, has reportedly exhibited relatively high Pb, Zn and Cu tolerance and may be considered a potential species for phytostabilisation purposes [4]. PTEs (Zn, Mn, Cu and Cd) were transferred from green to senesced leaves in *Cupressus sempervirens* [5]. Plantago major roots, suitable for phytoextraction from soil and phytostabilisation, facilitate the study of the efficacy of bioaccumulation for pollution with respect to PTEs [6]. Moreover, the soil–plant interactions in regulating the movement of PTEs from soil to plant can be useful in soil reclamation. Some plants, such as flax and hemp [7,8], can remove considerable quantities of PTEs from the soil with their root system. Mining activities, in particular, contribute to the increase in PTEs in soil and plants [9,10,11,12,13]. This is especially important when they are translocated from the soil to crops and then accumulate in them [14], causing serious human health concerns, even at low concentrations, due to their bioaccumulation abilities [15]. Diverse and productive vegetation is decisive for the success of soil reclamation [16]. The metal detoxification mechanisms of plants have been widely reported [17,18]. In addition, it is known that photosynthetic pigments in plants such as chlorophyll and carotenoids act to protect photosynthetic organisms under stress conditions [19,20], and proline (Pro) is a stress indicator in crops [21,22,23]. However, strategies to overcome the threats posed by metal stresses including antioxidant systems have been largely ignored in the area of reclamation [24]. A better understanding of the biogeochemical processes that control trace element cycling and the abundance of trace elements in habitats is likely to be key to their effective management and the diminishment of health risks due to these pollutants [25,26].

This study was based on the research concept of the circular economy, which is recognised as an effective method of neutralising industrial and municipal waste. A circular economy reduces the number of raw materials, waste, and energy consumed. It is achieved by creating a cycle (loops). Waste resulting from a technological process is reclaimed and reused as a raw material for other technological processes. The fractions are used as raw materials throughout the technological process to create new soil products. Overall, this contributes to a cleaner soil environment.

Our research utilised industrial and municipal waste, and we developed improved reclamation materials based on their physical properties. Wastewater treatment plants are an important component of a circular economy. In wastewater treatment plants, sewage sludge is formed through a variety of physical, chemical, and biological processes. Innovating technologies for sewage sludge processing improve the efficiency of the entire treatment process and make sludge a valuable raw material for other industries. Wastewater sludge should be regarded as a material (phosphorus, nitrogen) and energy resource. In this concept, sewage sludge is used as a raw material for further processing. Sludge from sewage treatment facilities can be a source of organic matter. As part of the waste product, coal waste consists of rock fragments that originated in layers within the seams. As clay minerals are a major component of coal waste, their physical and mechanical properties determine the soil properties and potential applications. As a soil substitute, we combined municipal sewage sludge with fly ash to enhance water permeability in degraded soils. As a stabiliser, this by-product was added to the waste substrate to prolong its retention time in the post-restored soil (extending the duration of a waste substrate is an important stage in the product life cycle). It is vital to keep the inspected soil product in circulation for as long as possible to implement a circular economy approach.

Low phosphorus and nitrogen levels are an additional problem in areas undergoing reclamation. Promoting the soil-forming process by adding organic materials with variable fertilising potential must be conducive to plant development (maintaining a permanent plant cover). By reusing the material, nutrients important for plant development can be recovered. The circular approach was associated with soil improvement in the recovered material, the soil substrate in the degraded area. In view of the close relationship between the biochemical features of plants and habitat conditions, we hypothesised that the application of soil substrate containing sewage sludge amendments to the reclaimed area could effectively improve the fitness of introduced vegetation and plant tolerance to PTEs. Therefore, we focused on: (1) adopting a circular economy approach using waste material in the reclamation process; (2) assessing physical and biochemical parameters using naturally occurring *Taraxacum officinale* in the initial soil (substrate); (3) evaluating the water deficiency response to plant growth along the contaminated area enriched with PTEs; and (4) determining the plant and soil substrate enzymatic activity after eight years of land reclamation.

## 2. Materials and Methods

### 2.1. Study Area and Experimental Design

The research was conducted in Zabrze, in an essential coal mining industrial area of the Upper Silesian Region (southern Poland). Soil substrate effectiveness in the area has degraded as a result of coal mine exploitation and processing, which began in May 2010. A coal seam and clay shales underlain sandstone layers and a calcareous substrate were the parent rock. The solid waste formation procedure consisted of the following elements: (a) post-flotation coal waste (25%), (b) municipal sewage sludge (35%), (c) crushed stone (25%); (d) fly ash (15%). A sub-structure of constructed sites comprising crushed stone (30 cm-thick) was produced, particularly for mine soil reclamation management to conduct remediation on mine heaps. Upon processing, granulated material was observed and according to the FAO definition, it was classified as sandy clay loam. Bioreclamation was initiated with soil substrate with a thin surface layer (50 cm-thick) for plant root establishment.

Three hundred Mg (tons) of soil substrate (biosolid material) was provided as an initial option for soil reclamation. On field experiments enriched with the soil substrate, deciduous and coniferous trees and shrubs were established by planting biennial seedlings. In 2010, the above-mentioned plots were planted with: *Larix decidua* Mill., *Pinus sylvestris* L., *Pinus nigra* J. F. Arnold (Pinaceae), *Populus tremula* L. (Salicaceae), *Betula verrucosa* Ehrh. syn. *B. pendula* Roth (Betulaceae), *Quercus rubra* (Fagaceae), *Prunus serotina* Ehrh., syn. *Padus serotina* (Ehrh.) Borkh. (Rosaceae), *Robinia pseudoacacia* L. (Leguminosae), *Acer negundo* L. (Aceraceae), *Tamarix parviflora* DC. (Tamaricaceae), *Elaeagnus angustifolia* L. (Elaeagnaceae), *Fraxinus excelsior* L. and *Hippophae rhamnoides* L. (Oleaceae). The assortment of plant species to be planted was determined on the basis of available peer-reviewed and published reports.

For the experiment, only plants that thrived in the post-extraction area were chosen, and they were the following tree species: *Pinus sylvestris*, *Salix alba*, *Acer negundo*, *Robinia pseudoacacia*, and among the shrubs *Elaeagnus angustifolia.* Herbaceous plants that grew spontaneously in the study area were included as well: *Taraxacum officinale* and *Tripleurospermum inodorum.*

During the experimental treatment of the plant material, the air temperature was within the range of 20–23 °C during the day and 15–17 °C at night, with plants receiving natural light which peaked at approximately 9000 µmol m^−2^ s^−1^.

### 2.2. Sample Collection

Eight years after initiating the described large-scale field treatment, selected plant samples were collected. In the case of plots on which shrubs and trees were grown, top soil substrate samples (0–20 cm) were taken using plot grids of 21 m × 7 m. Fifty soil substrate samples (550 cm^3^ soil volume) were collected from the root zone (20 cm) in each season. Organic debris was removed. Soil substrate samples were air-dried in heat-controlled ovens and sieved. Samples from plots of each replicate were placed in individual plastic bags and stored at 4 °C. All details and sampling procedures were described in a previous paper [27].

Additionally, for the biochemical activity assessment, we sampled the common dandelion (*Taraxacum officinale*) in each study plot (Figure 1a) and near the research area (Figure 1b; control site). The common dandelion (*Taraxacum officinale*) is a widespread plant which often exists in extremely polluted habitats. We selected *T. officinale* as the representative flowering perennial plant of the Asteraceae family with no habitat limitations and conservation problems as an indicator in the assessment of biochemical feature. *T. officinale* also has an excellent adaptability to low-fertility dry lands. Moreover, we assume that succession in the reclaimed area will be easier owing to the enrichment of the substrate with amendments. Plant samples, including the above-ground (shoot) and underground (root) parts, were taken as individual material. We stored *T. officinale* from both the study plot and one non-polluted outside (control site) plot near the research area. Forty samples were gathered (the leaves and roots were thoroughly mixed) in plastic bags between 15 April and 30 November (vegetation period in Poland) in each season. All performed field and laboratory studies were conducted according to soil reclamation procedures to measure physiological tolerance [28,29,30,31] with respect to exposure to stress factors for whole collected annual herbs during daylight time [32] in the derelict area.

### 2.3. An Investigation of Metal Concentrations in Soil

Wet digestion of dry substrate samples with concentrate acids (HCl and HNO_3_) in a microwave oven (Multiwave 3000, Anton Paar, Graz, Austria) was used to determine the Cr, Mn, Fe, Ni, Cu, Zn, Pb and Cd contents. The digestion was carried out in accordance with the programme using the following parameters: power 1400 W, temperature 240 °C, time to reach maximum power 5 min, time of maximum power 15 min, ventilation time 5 min and cooling time 40 min. The concentrations of elements were determined by using an inductively coupled plasma atomic emission spectrophotometer (ICP-OES) made by Perkin-Elmer. Substrate samples were analyzed in duplicate. Whenever the analysis results of the replications did not match by at least ±5%, an additional two analyses of the sample were conducted. In combination with the internal standard and the certified reference material CRM023-050—Trace Metals—Sandy Loam 7 (RT Corporation, Laramie, WY, USA), the quality of the determinations was verified using the Cr, Mn, Fe, Ni, Cu, Zn, Pb and Cd contents.

### 2.4. Concentrations of Metals in Plants

As part of the dry combustion method and using a 1:3 H_2_O:HNO_3_
*v/v* solution to dissolve the ash, the contents of Cr, Mn, Fe, Ni, Cu, Zn, Pb, and Cd in the plant material were determined. Extraction was performed with double distilled water. ICP-OES was used to determine the concentrations of elements.

### 2.5. Water Content Percentage Measurement

To determine water content (WC), leaf samples were weighed (FW), dried at 65 °C until constant weight (72 h) and then reweighed (DW); water content of each sample was calculated as:WC% = [(FW − DW)/FW] × 100.

Table 1 shows the physical properties of water in the tested soil substrate.

### 2.6. Analysis of Plant Biochemical Parameters

Five mg of lyophilised and homogenised sample were extracted in 1.5 mL of 95% ethanol for 15 min and centrifuged at 2000× *g *(Universal 32R, Hettich, Germany) for 10 min. The clear supernatant (100 μL) was transferred to a 96-well microplate and absorbance was measured at 470, 648 and 664 nm spectrophotometrically with a microplate reader (Synergy II; Bio-Tek, Winooski, VT, USA). The concentrations of total chlorophyll a + b(Chl) and carotenoids (Car) were calculated according to the equations given by Lichtenthaler and Buschmann [33], taking into account that the path length of microwells was filled to a quarter of their depth. Concentrations reported are averages of three biological replications, each consisting of two analytical replications, and the values were later converted to mg·g^−1^ DW.

Proline (Pro) content was quantified using dry leaf material according to the ninhydrin-acetic acid method of Bates et al. [34]. Pro was extracted in 3% aqueous sulphosalicylic acid and the sample was mixed with acid ninhydrin solution, incubated for 1 h at 95 °C, cooled on ice and then extracted with toluene. The absorbance of the supernatant was red at 520 nm using toluene as a blank. The Pro concentration was expressed as μmol g^−1^ DW.

Total flavonoids (TF) were measured in methanol extracts (0.1 g of dried material in 3 mL of 80% methanol) and determined following the method described by Zhishen et al. [35]; the absorbance was measured at 510 nm, and the amount of flavonoids was expressed in equivalents of catechin (mg C eq.·g^−1^ DW), used as standard.

Total phenolic compounds (TPC) were determined according to the modified method of Blainski et al. [29] using lyophilised plant material. A clear supernatant (50 μL) was diluted with 0.5 mL of deionised water and 0.2 mL of Folin–Ciocalteu reagent, and after 10 min 0.7 mL saturated Na_2_CO_3_ was added. The samples were incubated for 2 h, mixed and transferred to 96-well plates. Absorbance was measured at 765 nm on a micro-plate reader (Synergy 2, Bio-Tek Winooski, VT, USA). The level of total phenolics was expressed in equivalents of gallic acid (GA) (mg GA eq g^−1^ DW) used as standard.

### 2.7. Analyses of Enzymatic Activity

Incubation at 37 °C for 24 h with a 3,5,triphenyltetrazolium chloride solution as a substrate resulted in the measurement of dehydrogenase activity (DHA). The spectrophotometer (Shimadzu UV-1800, Kyoto, Japan) measured the wavelength of 450 nanometers. As a substrate, a sucrose solution was used to measure invertase activity (IA) after 24 h of incubation at 37 °C. With a Shimadzu UV-1800 spectrophotometer at 540 nm, the color intensity was measured. A two-hour incubation at 37 °C was followed by the measurement of urease activity. A urea solution was used as the substrate. Using a Kjeltec apparatus, the amount of ammonia remaining after urea hydrolysis was calculated. A soil dry matter value was calculated [36].

### 2.8. Soil-Substrate Physical Properties

By parametrising van Genuchten’s equations [37] and using water characteristic curves made for pressure chambers with porous ceramic plates, the water content was calculated. The Kopecky method was used to estimate bulk density and porosity [38].

### 2.9. Data Handling and Statistical Analysis

Principal component analysis (PCA) can reduce the set of *p* variables to the set of k variables, where k < *p*, with little information loss and few detected relationships between variables. Groups of parameters describing susceptibility to water stress and biochemical parameters were distinguished based on these multivariate statistics. PCA was used to demonstrate the correlation between biochemical parameters and physical and water properties in the soil substrate. The *p*-value of Bartlett’s test statistic indicates the correctness of a hypothesis concerning the existence of a significant difference between the obtained correlation matrix and the identity matrix. In this study, a correlation matrix was chosen to show the relationship. The Kaiser-Meyer-Olkin coefficient was used to check the degree of correlation between the original variables. This was applied to distinguish the strong evidence and relevance of conducting a PCA. The test was performed with PQstat version 1.8.2 (Poznań, Poland). A general regression analysis was applied to produce a model demonstrating the relationship between the proline and PTE content. The analyses were performed in the PAST statistical programme (version 4.03, Olso, Norway).

## 3. Results

Within the experimental research plot, the highest concentration of Zn was found. The metal Fe accumulated in the greatest quantity in all species, especially in herbaceous plants. Cd had the least accumulation and occurred, mainly in *Tripleurospermum inodorum*. Cu was primarily accumulated in *Taraxacum officinale*, Cr in *Acer negundo*, Mn in *Salix alba* and Pb in *Taraxacum officinale* and *Tripleurospermum inodorum* (Table 2).

In trees, the highest concentration was found for dehydrogenase. However, in the case of shrubs, the highest concentration was observed for urease in *E. angustifolia*. For herbaceous plants, *Tripleurospermum nodorumi* had a higher concentration of all enzymes tested compared to *Taraxacum officinale* in study plots (Table 3). The physiological features for *Taraxacum officinale* were similar in both plots (Table 4). Multivariate regression for the dependent variable (Pro) showed statistically significant correlations for TPC, Mn, Zn, Ni and Cd (Table 5).

Bulk density is a significant factor in the assessment of reclamation effects and considerably affected the value of the infiltration coefficient after only four years of reclamation of sandy mining areas. Furthermore, the first effects were visible one year after introducing the plant species. A negative correlation was found between the bulk density and the total porosity (Figure 2). The Mn content was statistically significant with the increase in Pro (Figure 3). A PCA was used to determine the relationship in *T. officinale*, by adjusting the physical-water parameters and physiological properties. The total porosity (TP) was related to the content of Pro and chlorophyll a (Chl-a). The bulk density (BD) was internally correlated with the total phenolics (TF) content. For the content of carotenoids (Caro), a very close relationship was found with the total content of flavonoids (TPC). The (MWC) showed a strong negative correlation for the two axes of the factors (Figure 4). Based on the average water content of the leaves, we found the following:*Pinus sylvestris*—43.30%;*Salix alba*—34.57%;*Acer negundo*—46.71%;*Robinia pseudoacacia*—38.16%;*Elaeagnus angustifolia*—41.82%;*Taraxacum officinale*—67.09%;*Tripleurospermum inodorum*—70.20%.

## 4. Discussion

### 4.1. Assessment of the Phytomelioration Approach to Reclamation

Plants applied for remediation purposes can spontaneously grow on sewage sludge substrate in abandoned areas [39]. Sewage sludge can be used in reclamation of contaminated areas and may be applied as an immobilising agent for phytostabilisation purposes [40]. It has been shown that plants growing on experimental plots may have sufficient water to ensure their proper development or suffer from water shortage due to other factors [27].

Waste such as sewage sludge, post-flotation lime, and mineral wool provide nutrients for plants and soil microorganisms and can be utilised to enhance the quality of soil [41]. The vegetation cover in degraded areas is well known for its many functions that are necessary in the reclamation process [42,43,44]. These functions include stabilisation and neutralisation of pollutants in the substrate and extraction and storage of PTEs [45,46].

The lack of water leads to severe plant stress in drought-affected regions [47]. Water stress is measured by certain indicators, such as relative water content, photosynthetic pigments, and proline levels. Both relative water content and chlorophyll level have been correlated with drought tolerance [48]. With a reduction in transpiration rate, plants lose water as they become dehydrated. In response to drought stress, plants increase the expression of stress-related genes in aerial parts, including the genes involved in proline metabolism [49]. Salinity and drought lead to plants responding primarily through Pro accumulation, which regulates strategies to combat stress in agricultural production [21]. Chlorophyll can reflect the nutrition status of vegetation by providing an indication of vegetation stress and external disturbance [50]. Under drought stress, the chlorophyll content/components of the plants are usually altered, leading to a compromised photosynthetic apparatus [51]. Drought triggers a reduction in chlorophyll content, and Pro content increases many times under water stress [47]. The total porosity was correlated with both chlorophyll and Pro content, and bulk density was related to total phenolic content in this study.

The current set of measurable criteria for assessing the quality of reclamation works and classification criteria for emerging soils and habitats should be improved by adding a reliable index reflecting degradation of post-mining areas. Dynamic growth analyses of plant communities have been used to assess natural succession in degraded areas [52,53].

### 4.2. Physiological Response of T. officinale to Abiotic Stress

A recent study showed that the regulation mechanism of *T. officinale* contains a wide range of remarkable bioactive substances [54] and may affect metal absorption in municipal solid waste landfill vegetation [55]. *T. officinale* leaves growing spontaneously in grasslands and along roads are traditionally picked and eaten in some countries. The above-ground vegetative parts of *T. officinale* have been found to offer a superior source of nutrition compared to other perennial or annual plants [56]. *Taraxacum* sp. endure in stressful conditions due to high phenolic compound concentrations which effectively suppress the oxidative stress induced by a high level of risk elements in urban soils [57]. Bretzel et al. [58] found that increasing flavonoid content under stress conditions influenced the risk of contaminated sites. Measuring antioxidants is essential for the identification of stress factors [59]. Species used in this study accumulated PTEs from post-mining soil with high efficiency due to their metabolic function. For all the presented reasons, the common dandelion (*T. officinale*) seems to be an interesting plant for studying the process of accumulation of anthropogenic pollution, particularly PTEs [60]. Malizia et al. [61] indicated that PTEs accumulated at high concentrations may impair plant physiology by reducing growth or respiration, restricting photosynthetic processes and inhibiting fundamental enzymatic reactions. Application of waste to degraded soil induced changes not only in the number of microbial groups, but also in their respiratory and enzymatic activities [41]. Enzymatic activity is one of the parameters used in soil monitoring and soil quality assessment [62]. This connection was revealed using a multivariate regression model. The multivariate regression model highlighted the relationship of Pro with Mn r > 0.65; *p* < 0.001 and Fer = −0.53 (*p* < 0.01). As Pro increased, the content of Mn increased, and Fe decreased (Table 5). The PCA showed a strong relationship between chlorophyll and Pro in common dandelion with capillary porosity in the soil material. Carotenoids were linked to TPC (Figure 4). When plants absorb certain PTEs ions from soil in a mining area, their chlorophyll content decreases. As a result of metal stress, photosynthetic pigments are changed, indicating that the photosynthetic apparatus is damaged and the photosynthetic capacity of the plant is diminished. A plant exposed to Hg at a concentration of 10 mg kg^−1^ had chl a and b decrease of over 40% and 48%, respectively, in comparison with a blank sample. Studies on tobacco plants have demonstrated that Cd stress inhibits the expression of key enzymes during chlorophyll synthesis, resulting in a decrease in chlorophyll content. However, Zn stress did not affect chlorophyll concentration significantly [63].

Fluctuation in PTEs may be caused by non-enzymatic antioxidants (carotenoids and Pro) in crops such as red cabbage. A soil amendment, however, causes chl a and chl b levels to be reduced significantly. Additionally, soil amendments changed total carotenoid concentrations in plants [64]. In the case of *Arabidopsis thaliana*, generally increasing metal concentrations resulted in a gradual decrease in chlorophylls and a fluctuation in carotenoid content, depending on the metal type [65]. More plant growth and chlorophyll, carotenoid and Pro contents were found in carrots grown in fly ash mix soil than in plants grown in soil without amendments [66]. Other soil amendments such as biochar enhanced chlorophyll a and b, total chlorophyll and carotene by 10, 45, 17 and 15%, respectively, in tomato plants [20]. Soil amendments with arbuscular mycorrhizal fungi improved chlorophyll a and total chlorophyll contents by 31–35% and 60–75%, respectively [67].

The metal-induced stress in plants provokes changes such as the synthesis of compatible molecules (osmolytes). Pro accumulation is one of the most general responses of plants, as it is the major functional osmolyte in many species. Biosynthesis of Pro accelerates when plants have toxic metal content, helping to protect them. This makes Pro one of the most effective compatible molecules produced under adverse conditions for helping the plant tolerate stress, including metal toxicity [68,69]. In *Trapa natans*, the Zn content stimulates the greatest accumulation of Pro in roots, in comparison to Cu, Cd and Pb [70]. In *Capsicum annuum* L., soil amendment of a K-rich carrot compost under drought significantly modified plant metabolic activities, including the level of free Pro [71].

Soil amendments, such as magnetite, decreased Pro concentration in mandarin trees under salt stress conditions. Soil amendments with arbuscular mycorrhizal fungi, selenium, AMF and Si-gel enhanced chlorophyll a and total chlorophyll contents in peas under high arsenic levels, whereas the Pro level was lower than that of the control under the same conditions [67]. In species such as Abies alba, a clear connection was observed between chlorophyll content and antioxidant activity; the more chlorophyll, the more antioxidants [72]. Our model revealed that Pro was the most suitable among the biochemical biomarkers. This indicates that *Taraxacum officinale* can be used in biomonitoring PTEs using Pro and chlorophyll as sensitive biomarkers (Table 3, Figure 4). Pro is another of the most accurate water stress indicators. In this study, it was found that the addition of waste substrate to the studied area improved both plant resistance to PTEs excesses and drought resistance. Abiotic factors restrict tree growth, making tree species vulnerable to extinction. When choosing plant sites on brownfields, plant–soil relationships should be considered. In herb plants, water was accumulated in leaves at a rate of more than 70%. *Acer negundo* accumulated the most water of all trees studied. *Elaeagnus angustifolia* leaves have absorbed 41.82% of water. This suggests that leaf water is effectively captured.

### 4.3. Potentially Toxic Elements in Degraded Areas

Several sources of PTEs are found in domestic, industrial, agricultural, medical, and technological applications, leading to their wide distribution in the environment [73,74]. Thus, PTEs contamination is one of the most common sources of environmental pollution. In the present study, PTEs were found to differ significantly between plant species. Therefore, species with higher metal concentrations should be monitored more frequently. The *Acer negundo*, proved to be the most appropriate biomonitor for Cr, while *Taraxacum officinale* was used for Cu and *Salix alba* for Mn. Vascular plants respond to their accumulation physically and biochemically [75]. High values of PTEs were observed in the soil and dandelion sampled, in a locality with high traffic density situated in the Czech Republic [76]. Zarzycki and Petryk [77] indicated that influencing the accumulation of PTEs in plants is determined by different factors which depend on the metal types and main soil parameters. According to their results, factors affecting bioaccumulation of PTEs included organic matter and soil reactions for cadmium, organic matter and electrical conductivity for lead and pH for copper. Variation in accumulation of Cr, Fe, Ni and Cu has been documented in tree species [78]. Therefore, many tree species are used as PTEs indicators, even in industrial areas [79]. It is important to determine the concentrations of PTEs contained in leaves. The metal accumulation index is a useful measurement for urban plant leaves that indicates emissions and coal combustion [80,81]. We assumed that this species could accumulate PTEs from post-mining soil with more efficiency than other studied plants.

In the case of *Miscanthus giganteus*, the use of sewage sludge as a substrate for cultivation resulted in a three times higher consumption of Ni, which was twice as high as the accumulation of Cr and Cu, and increased accumulation of Zn and Cd in the third year of cultivation [82]. Therefore, the analysis of the chemical composition of vegetation is required in reclamation areas [83,84]. Energy plants are characterised by their high ability to uptake PTEs from substrate. The use of sewage sludge contributes significantly to metal bioavailability [25]. The high content of Zn and Pb in the substrate in turn favours the emergence of vascular plants, which are clearly distinguished by different physiognomy and composition from other communities and are often referred to in the literature as galmanic communities.

When soil is formed and mined, PTEs are released from parent rocks into the soil. Pb can also accumulate in soil from nearby communication routes. With the help of extensive assimilation systems, dust from industrial areas is deposited on leaf blades in areas with industrial emissions. Leaf tissue is only partially penetrated by surface-deposited PTEs and dust. Thus, *Taraxacum officinale* and *Tripleurospermum inodorum*, which have an extensive root system, contained the most Pb (Table 2).

The accumulation of metal in leaf blades may be caused by damage from frost or other environmental factors. Additionally, the metal-bearing dust clouds penetrate through the wax layer on the leaf surface. The accumulation of metals occurs in both trees and herbaceous plants. Cd is absorbed and transported efficiently through roots to all organs by plants. Plants with a well-developed root system grown in degraded soils are particularly susceptible to the absorption and exposure of Cd.

Zn and Cu accumulate differently, and no relationship has been found. Growing season and cultivation time were also major factors in the experimental reclamation plot. Plant species with a short growing season, especially those planted in early spring, were more contaminated, according to our observation. An explanation for the high exposure of trees to contamination is difficult to find.

In spring, elevated temperatures and moist soil increase the mobility of PTEs within the soil subjected to reclamation. Possibly, this is due to the increased enzyme activity. This can be seen in the highest concentration of dehydrogenase and urease (Table 3). Plants rely on urease activity to replenish nitrogen after proteins are broken down into urea. The displacement of PTEs may also be affected by urease activity.

A PCA showed a negative correlation with mass water content and a positive correlation between chlorophyll a content and capillary porosity (Figure 4). Since post-exploitation soils accumulate rainwater, the reclaimed soil impact is high. The presence of rainwater may impede the physiological plant process and barriers for metal accumulation. Additionally, rainwater imposes the leaching of PTEs into the soil profile.

According to our conclusion, post-industrial waste can regulate properties along soil–plant pathways, which makes it an appropriate substrate. This waste substrate had a high accumulation of PTEs, indicating its suitability for phytoremediation. PTEs can be captured in waste substrate, preventing their migration to plants.

## 5. Conclusions

Photosynthetic pigments (chlorophyll a and b, and carotenoids), Pro, antioxidants (total flavonoids) and total phenolic compounds are suitable bioindicators of the physiological responses of plants under stress conditions due to high PTEs concentrations and for assessing the effects of restored areas ameliorated by sewage sludge amendments. Chlorophyll was a better indicator in this study compared to carotenoids, which can be attributed to their lower vulnerability to the negative impact of PTEs. Pro indicated that the amendments stimulated an improvement in the physiological responses of the plants to increase their tolerance to PTEs. Total flavonoids and total phenolic compounds were higher in plants located at the control site as a response to metal stress. In this study, the biochemical activity of plants is proposed to fully access their impact on substrate soils (mobility, bioavailability and ecotoxicity). The interpretation of the results indicates directions for future research, including the assessment of the concentration of reinforcing elements. There is a close relationship between the physiological and biochemical features of plants and the soil or substrate where they are growing. The results may provide new insights into abiotic post-mine plant growth limitations, such as salinity stress, including oxidative stress, as the effect of stress factors (regarding biochemical activity), and provide a baseline reference for rational restoration of mining areas. The application of soil substrate containing sewage sludge amendments in the reclaimed area could effectively improve the fitness of introduced vegetation and the stress tolerance to PTEs of plants; therefore, soil substrate is suitable for circular land reclamation.

## Figures and Tables

**Figure 1 ijerph-19-05296-f001:**
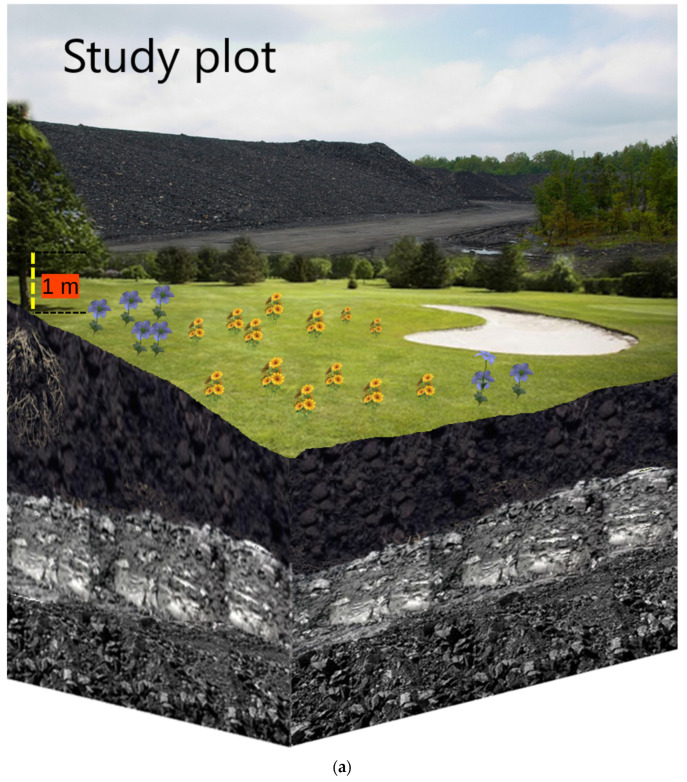
(**a**) Experimental research field located near topsoil heaps in a degraded area. (**b**) Control plot adjacent to the degraded area with spontaneous vegetation.

**Figure 2 ijerph-19-05296-f002:**
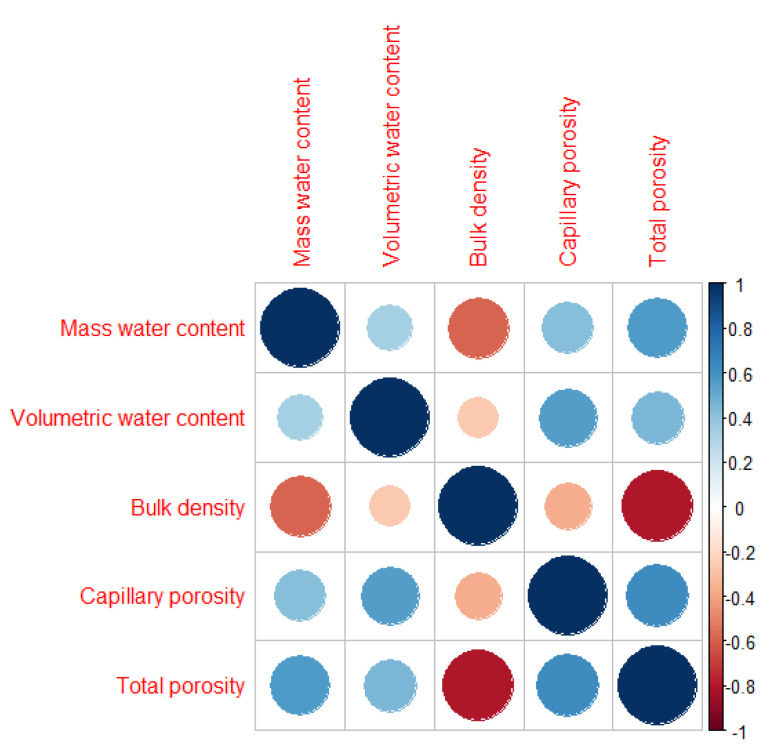
Spearman correlation for soil water parameters.

**Figure 3 ijerph-19-05296-f003:**
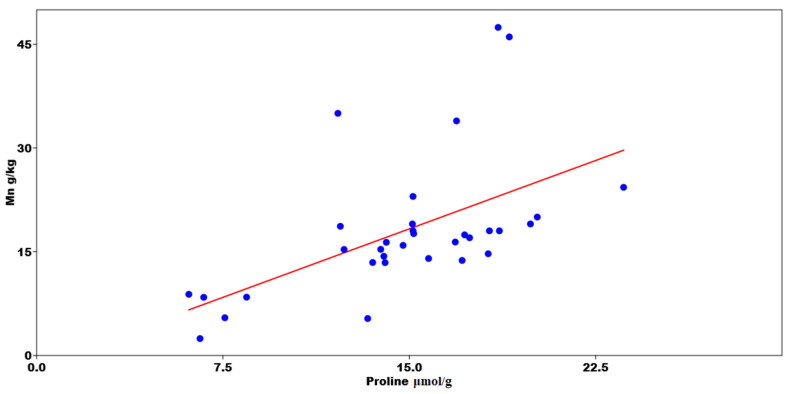
Results of multivariate regression between proline (Pro) concentration and Mn content.

**Figure 4 ijerph-19-05296-f004:**
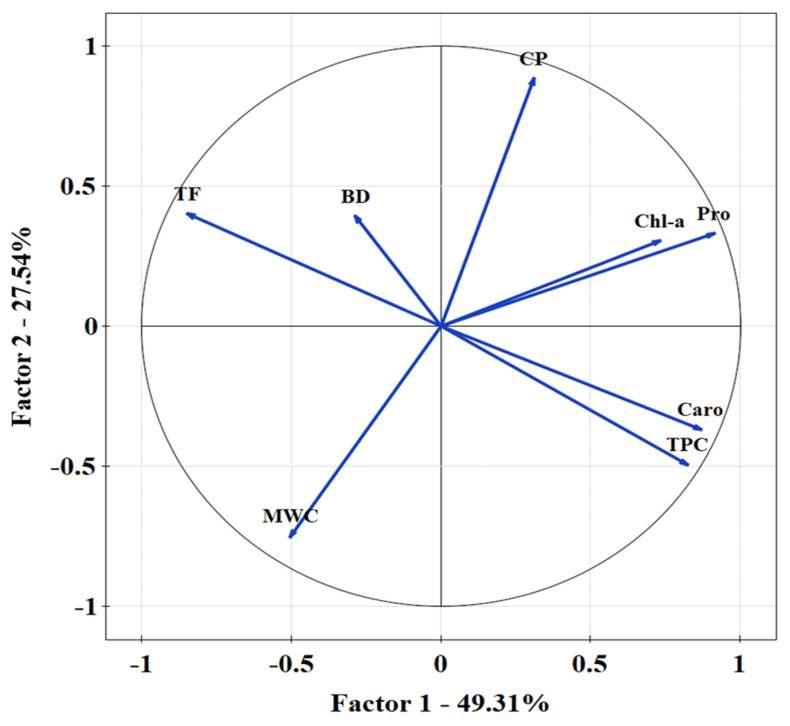
Principal component analysis for water parameters in soil substrate and biochemical properties measured in *T. officinale*. Kaiser-Mayer-Olkin ratio = 0.704; Barlett’s test *p* < 0.001.

**Table 1 ijerph-19-05296-t001:** Water and physical properties of the studied soil substrate (data from Halecki and Klatka, 2018).

Layer	Mass Water Content	Bulk Density	Volumetric Water Content	Total Porosity	Capillary Porosity	Soil Moisture	Organic Mater Content	Soil Texture
cm	g·g^−1^	g·cm^−3^	cm^−3^·cm^−3^	%	%	%	%	
10	0.41	1.54	0.45	44.11	38.40	36.35	28.99	sandy clay loam
20	0.37	1.47	0.48	48.17	37.61	30.25	29.53	sandy clay loam
30	0.45	1.55	0.49	49.30	39.84	23.32	25.50	sandy clay loam
50	0.43	1.57	0.52	49.18	40.00	20.41	23.00	sandy clay loam

**Table 2 ijerph-19-05296-t002:** The content of PTEs in the soil of the replanted area, as well as on the harvested plant parts.

PTEs	Cr	Ni	Cu	Zn	Cd	Pb	Mn	Fe
	Soil Substrate							
*Pinus sylvestris*	45.24	76.93	36.65	177.91	5.86	84.10	78.53	13.34
*Salix alba*	38.76	34.86	45.53	431.02	6.10	98.54	41.23	14.40
*Betula pendula*	28.45	51.32	29.02	384.21	5.03	74.84	35.12	27.21
*Acer negundo*	78.89	60.38	91.25	466.92	4.25	141.13	34.67	26.08
*Robinia pseudoacacia*	41.63	45.89	78.67	343.09	7.90	97.05	35.45	17.65
*E. angustifolia*	53.55	54.54	64.55	355.54	8.69	36.45	61.95	16.43
*Taraxacum officinale*	53.63	43.25	23.95	278.30	2.94	78.76	0.87	20.76
*Tripleurospermum inodorum*	45.54	35.75	26.97	357.76	3.98	98.08	0.90	18.98
	Plants							
*Pinus sylvestris*	4.54	17.45	3.46	100.49	0.89	13.90	46.78	0.14
*Salix alba*	22.65	45.67	2.54	366.6	0.48	5.62	78.34	0.65
*Betula pendula*	12.45	19.41	6.32	98.14	0.61	1.43	73.14	0.24
*Acer negundo*	45.54	1.63	9.43	32.43	0.54	5.74	26.45	0.74
*Robinia pseudoacacia*	2.54	3.65	7.46	26.65	0.75	0.57	12.53	0.45
*E. angustifolia*	4.12	1.67	12.45	28.56	0.98	4.94	41.05	0.46
*Taraxacum officinale*	12.43	34.54	23.94	84.93	0.57	34.65	0.46	35.54
*Tripleurospermum inodorum*	13.87	24.76	16.76	89.97	3.86	35.87	0.57	27.76

**Table 3 ijerph-19-05296-t003:** Mean content of enzymes in plants and soil substrate.

Species	Invertase (mg Invertasied Sugar kg Soil/24 h)	Urease (mg N-NH_4_/kg/2 h)	Dehydrogenase (mg TPF/kg Soil/24 h)
*Pinus sylvestris*	0.14	77.98	289.69
*Betula pendula*	0.09	90.84	54.7
*R. pseudoacacia*	0.01	36.75	12.76
*E. angustifolia*	0.36	40.35	36.97
*Taraxacum officinale*	0.21	35.21	13.43
*Tripleurospermum inodorum*	1.35	86.39	17.76
Soil substrate	0.23	52.92	13.01

**Table 4 ijerph-19-05296-t004:** Range of photosynthetic pigments (chlorophyll a, chlorophyll b and carotenoids), proline concentration, total flavonoids (TF) and total phenolic compound (TPC) in *T. officinale* (control site and studied plot) during vegetation period.

Plant Species	Chlorophyll a	Chlorophyll b	Carotenoids	Proline	TF	TPC
	(mg/g) DW	(mg/g) DW	(mg/g) DW	(µmol/g) DW	(mg/eqC/g) DW	(mg/eq GA/g) DW
*T. officinale*	15.08–43.88 (31.79) *	3.25–16.14 (10.17)	53.09–461.49 (215.56)	15.76–20.15 (17.64)	1.93–2.52 (2.32)	3.29–4.68 (3.89)
*T. officinale* (control site)	10.97–29.72 (23.51)	3.17–10.16 (5.12)	55.43–914.37 (354.74)	7.58–17.22 (13.14)	0.27 –2.95 (1.32)	3.03–5.54 (3.82)

*—Mean values are in parentheses.

**Table 5 ijerph-19-05296-t005:** Multivariate regression for proline as the dependent variable and biochemical parameters and PTEs.

Variable	Slope	Error	Intercept	Error	r	*p*
Carotenoids	−3.69	2.77	204.05	62.43	−0.23	0.19
Chlorophyll a	0.03	0.02	1.88	0.45	0.24	0.17
Chlorophyll b	−0.13	0.06	6.86	1.43	−0.34	0.05
TPC	−0.39	0.18	21.83	4.16	−0.35	0.04
TF	−0.02	0.02	2.43	0.35	−0.27	0.12
Mn	1.06	0.25	−7.66	5.58	0.62	0.03
Cr	−1.69	2.67	110.73	6.43	−0.11	0.53
Fe	−0.43	0.12	1.65	2.73	−0.54	0.32
Ni	−0.71	0.33	3.63	7.36	−0.36	0.04
Cu	−1.27	0.36	58.42	8.04	−0.53	0.32
Zn	−3.91	1.67	238.83	37.69	−0.38	0.03
Cd	−0.11	0.04	4.97	0.85	−0.46	0.01
Pb	−2.06	0.55	81.52	12.34	−0.55	0.32

## Data Availability

No new data were created or analyzed in this study. Data sharing is not applicable to this article.

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
