# Peer review of "A Circular Economy Approach to Restoring Soil Substrate Ameliorated by Sewage Sludge with Amendments"

_ijerph, 2022, doi:10.3390/ijerph19095296_

Round 1
Reviewer 1 Report
The matter of the manuscript ‘A circular economy approach to restoring soil substrate ameliorated by sewage sludge with amendments’ is noteworthy and fits into the scope of the International Journal of Environmental Research and Public Health. The presented study is appropriate and thought-provoking, which, in general, deserves publication. Still, the current version of the manuscript requires revisions and additions. I would like to point out the issues I am concerned about.
Key questions
- Let’s start with the title. I understand that this is a clickbait phrasing: “A circular economy approach…”. Chances are that more people will then learn about the ways we reutilize the sewage sludge. Actually, some of the general public may learn about it for the first time, and this is a good result per se. Nevertheless, if the title starts with these words, the approach should be discussed in detail, so everybody can understand why soil amelioration is a great initiative and how it can be a true part of the recycling chain.
- Speaking deeper about the study object, I want to say that it would be beneficial if the manuscript had a proper (or even short) description of technological processes in which the sludge is treated. As we know, this waste type is usually rather hazardous and thus requires special attention.
- The first part of the discussion on soil-plant nexus (l. 441-457) consists mainly of summaries from other works. When it finally comes to the words about the case study (l. 458-467), the paragraph is very formal and few numbers from Table 1 are interpreted. The reasons and barriers for metal accumulation or mobilization are poorly discussed. The analysis of how soil metal levels influence the translocation also leaves much to be desired.
- Furthermore, Table 1 itself looks unusual: the section “Soil substrate” includes biological species (Pinus sylvestris, Salix alba, etc.) and gives no idea of soil properties (e.g., sand/loam, moisture level, content of organic carbon, etc.). When explaining the soil processes, it is compulsory to mention the background natural conditions. So please provide more data on a) soil types of the study area; b) soil types and parent material of undisturbed adjoining areas. I suggest using the FAO soil classification. By understanding waste rocks and natural soils, we will be able to judge, whether the excesses are human-related, or rise from the natural conditions. That will be important for understanding the local transformation processes. Maybe the authors find relevant studies in the recently published special issue (https://www.mdpi.com/journal/minerals/special_issues/AMA_II).
- Please avoid the use of the term ‘Heavy metal’ throughout the manuscript. Please correct throughout the manuscript. According to the IUPAC recommendation (Duffus, 2002) the use of the term ‘heavy metal’ should be avoided, because this term is generally frowned upon these days, especially by environmental chemists, which consider the term meaningless and misleading. ‘Potentially hazardous elements’ or ‘Potentially toxic elements’ should be used. It can also be used Metals, ‘Metals/metalloids’ if arsenic and antimony are also studied as these elements are not metals. (Duffus, J.H., 2002. "Heavy metals" — a meaningless term? IUPAC technical report. International union of pure and applied chemistry 74, 793-807. || Hodson, M.E., 2004. "Heavy metals—geochemical bogey men?" Environmental Pollution 129, 341-343.).
Minor remarks
- The last paragraph of the Introduction chapter is more like the Conclusions (“The results…”, l. 83-91). Please consider rewriting it or relocating in the text.
- 1a — which part is the real photo and which is a digital model? As for Fig.1b, everything seems to be a render.
- 1a,b — please provide the scale and descriptions of mine waste layers and species depicted
- Please make sure that you either use Oxford comma or don't use it: “Zn, Mn, Cu and Cd” (l. 50) / “Ni, Cu, Zn, Pb, and Cd” (l. 177).
- Please do not forget that in botany, italic is generally used for the Latin binomial names of plants, as sometimes it is missed (see l.51).
I honestly hope you will find my suggestions supportive and wish you good luck with the paper.
Kind regards,
Reviewer
Author Response
We, authors, appreciate the editor and reviewers for careful evaluation of our manuscript. We also appreciate their time and efforts for valuable feedback to improve our manuscript. We have incorporated most of the suggestions with careful consideration fitting our research aim. Those changes are highlighted in the revised manuscript. Our answers to the reviewers’ comments and concerns are shown below in a point-by-point response. All page numbers refer to the revised manuscript with tracked changes. The manuscript has been proofread thoroughly, and we sincerely hope it will meet with your approval.

Reviewer 2 Report
The manuscript focuses on the biochemical features of trees, shrubs, and herbaceous species, growing for eight years on a soil substrate, that contains sewage sludge amendments. The study is interesting and well-structured. In particular, the study derives from an experiment carried out in previously published articles. I suggest better introducing the main results of the previously published articles in the introduction as well as in the material and methods, specifically about the experimental design and the properties of initial soil with sewage sludge amendments.
Below I reported some comments and suggestions:
Abstract.
Lines 23-34. I suggest modifying this section, reporting the main results of the experiment and briefly the conclusion.
Material and Methods.
Lines 98-99. Please specify the composition of the soil substrate.
Lines 107-112. There is a list of species used in the experiment, however, only the results of some species are reported. Please specify which species are discussed in the manuscript, including Tripleurospermum inodorum, dividing the species into trees, shrubs and herbaceous plants.
Lines 136-139. Please move this sentence in the introduction.
Line 195. The results about water content percentage are not reported.
Results
Lines 270-271. “There was a greater quantity of the metal Zn and a lesser quantity of Mn”. Do you mean “than the other PTE in the same plot”? Please specify.
Lines 272-275. Is the comparison of PTE contents amongst species or types of plant (tree, shrubs or herbaceous plants)? Please specify, dividing the species in table 1 according to each type. I suggest also reporting statistical analysis.
Line 275. Betula pendula is not reported in Tables 1 and 2.
Lines 286-291. As mentioned above, I suggest reporting a statistical analysis in order to support your results. Do you compare the enzyme content amongst types of plants (Tree, shrubs or herbaceous plants)? Please specify, dividing the species in table 2.
Lines 292. Please specify which kind of plots, e.g., study plot and non-polluted outside.
Line 292. Does table 4 report the same data mentioned in the previously published article? If yes, the data are different, and I suggest checking them and moving the table in the M&M section, as characterization of soil substrate. Otherwise, If the physical properties of soil substrate have been recently measured, please specify.
Discussion
In general, the discussion should be improved. The main results are listed in the conclusion section, and I suggest moving lines 469 and 488 to the discussion, comparing the results with published data.
Line 351. The data about water content are missing, please provide them. However, the relative water content (RWC) is reported in the discussion as an indicator of water stress and the determination of RWC is different from the water content proposed. The RWC is defined as the amount of water in a leaf at the time of sampling relative to the maximal water that a leaf can hold, after a re-hydration. Please check and eventually modify.
Author Response
Hereby we submit for publication a revised and supplemented version of our manuscript. We are very grateful for all your comments and valuable suggestions. We would also like to thank the Reviewer for his/her recommendations and thorough review of the manuscript. The manuscript was carefully checked for accuracy and all comments were adhered to. Please find below our responses to the Reviewers’ comments. References were formatted according to the journal style. We address all suggestions in detail below. We have marked all corrections in the file “Manuscript changes marked”.

Round 2
Reviewer 1 Report
Thank yo for your answers. I recommend that the MS is accepted.
Fig. 1a, 1.b – I suggest indicating "1 m" instead of simply "m".
Please check the typos like "Postgraduate Collage of Agricultural Sciences, Montecillo Campus" – it should be "College".
Author Response
We appreciate your time and consideration. Unfortunately, we did not take notice of it. Thanks.
We've also corrected Figure 1.
Reviewer 2 Report
The authors have improved the manuscript, however the values reported in table 1 seem to be different from those reported in Halecki and Klatka (2018), please verify and eventually correct the data. In addition, to improve the discussion, I suggest moving lines 586 and 602 to the discussion, comparing the results with cited articles.
Author Response
When rewriting the data, we overlooked this. Values for the table were added. Please accept my appreciation for this review. The discussion has been improved. Discussion chapters with added text are marked in red.